# A Brief Journey through Protein Misfolding in Transthyretin Amyloidosis (ATTR Amyloidosis)

**DOI:** 10.3390/ijms222313158

**Published:** 2021-12-06

**Authors:** Alejandra Gonzalez-Duarte, Alfredo Ulloa-Aguirre

**Affiliations:** 1Departamento de Neurología, Instituto Nacional de Ciencias Médicas y Nutrición Salvador Zubirán, Vasco de Quiroga 15, Col. Belisario Dominguez Sección XV, Tlalpan, Mexico City 14080, Mexico; 2Red de Apoyo a la Investigación, Instituto Nacional de Ciencias Médicas y Nutrición Salvador Zubirán, Coordinación de la Investigación Científica, Universidad Nacional Autónoma de México, Mexico City 14080, Mexico; aulloaa@unam.mx

**Keywords:** TTR amyloidosis, hereditary amyloidosis, protein misfolding, oligomer toxicity, ATTR variants, wild-type amyloidosis

## Abstract

Transthyretin (TTR) amyloidogenesis involves the formation, aggregation, and deposition of amyloid fibrils from tetrameric TTR in different organs and tissues. While the result of amyloidoses is the accumulation of amyloid fibrils resulting in end-organ damage, the nature, and sequence of the molecular causes leading to amyloidosis may differ between the different variants. In addition, fibril accumulation and toxicity vary between different mutations. Structural changes in amyloidogenic TTR have been difficult to identify through X-ray crystallography; but nuclear magnetic resonance spectroscopy has revealed different chemical shifts in the backbone structure of mutated and wild-type TTR, resulting in diverse responses to the cellular conditions or proteolytic stress. Toxic mechanisms of TTR amyloidosis have different effects on different tissues. Therapeutic approaches have evolved from orthotopic liver transplants to novel disease-modifying therapies that stabilize TTR tetramers and gene-silencing agents like small interfering RNA and antisense oligonucleotide therapies. The underlying molecular mechanisms of the different TTR variants could be responsible for the tropisms to specific organs, the age at onset, treatment responses, or disparities in the prognosis.

## 1. Introduction

Transthyretin (TTR) is a protein produced in the liver and choroid plexus that carries thyroxine and holo-retinol binding protein in serum [1,2]. It is an abundant protein in serum and cerebrospinal fluid (CSF). ATTR amyloidosis is caused by the deposition of insoluble amyloid fibrils in extracellular tissues of most organs, including the ones that may not display signs of disease [3]. ATTR amyloidosis comprises wild-type (WT) amyloidosis (ATTRWT) and hereditary mutated variants amyloidosis (ATTRv). ATTRWT classically is present with cardiomyopathy, carpal tunnel syndrome, and radiculopathy in older patients, usually above the age of 70 years. ATTRv amyloidosis is seen in younger (early-onset) and older patients (late-onset), resulting in peripheral neuropathy, cardiomyopathy, or a mixed combination [1,2]. The phenotypes vary depending on the mutation [2].

Pathogenesis involves the formation, aggregation, and deposition of amyloid fibrils in different organs and tissues [2]. However, other toxic mechanisms of TTR amyloidosis have recently been described. Therapeutic approaches have evolved from orthotopic liver transplants to novel disease-modifying therapies that stabilize TTR tetramers, and gene-silencing agents like small interfering RNA and antisense oligonucleotide therapies [1]. As new disease-modifying therapies emerge, a complete understanding of the molecular mechanisms subserving the development of TTR amyloidosis will provide helpful insights into the heterogenic manifestations of this misfolding disease. This knowledge could also aid in designing new therapeutic strategies and drugs or implementing tailored approaches for specific subpopulations of patients suffering from TTR amyloidosis.

## 2. Structure of TTR

TTR is a 55 kDa tetramer composed of 127 amino-acid subunits exhibiting an unusually high β-sheet content [4]. It possesses a conserved structural prototype of four identical monomeric subunits that form a central channel or binding site (Figure 1). The tertiary structure is composed of residues organized into eight β-strands identified by the letters A to H, connected by loops (Figure 1a,b). The strands C-B-E-F are oriented orthogonally to the strands D-A-G-H (Figure 1b). Eight β-strands conforming two β-sheets, stack with each other to establish a β-sandwich tertiary structure, with a short a α-helix between the β-strands (Figure 1b). The quaternary structure is characterized by the presence of four identical protomers (Figure 1d) [4,5], giving rise to two hydrophobic pockets for binding T4 [6]. The binding pockets for T4 are created by dimer-to-dimer interactions: one dimeric interface is formed between the subunits A and C, and B and D, sustained by an extensive hydrogen-bond network involving the β-strands F and H, whereas the other dimeric interface is formed between the subunits A and B as well as C and D, and stabilized by hydrophobic interactions [5]. The tertiary and quaternary stability of TTR correlates with its amyloidogenic propensity.

TTR-originated amyloid fibrils have two different morphologies type A and type B. Type B is composed of full-length TTR and has relatively long fibrils arranged in parallel bundles, whereas type A amyloid possibly is a truncated protein consisting of short fibrils tightly packed and disoriented within the carboxyl-terminal fragment of TTR [7]. The association of the type of amyloid with amyloidosis variants is less well understood. However, all organs of an individual patient contain ATTR deposits of either type A or B fibrils, and the type of amyloid deposit remains unchanged over time [3]. Type B TTR is the main component in patients with ATTRv amyloidosis, whereas type A fibrils are found in ATTRv and WT amyloidosis [3]. So far, only for 29 mutations, the type of fibril composition has been examined [3]. Amyloid fibril composition may be related to age at onset or pattern of inheritance (Table 1).

## 3. Protein Misfolding, Toxicity, and ATTR Amyloidosis

Protein misfolding, like that exhibited by mutant TTR proteins, is a process that can occur due to genetic, environmental, or sporadic factors. ATTR amyloidosis belongs to a group of diseases commonly referred to as proteinopathies or protein-misfolding diseases. More than 130 pathogenic mutations of the TTR gene have been described to date [1]. While the result of amyloidoses is the accumulation of amyloid fibrils resulting in end-organ damage, the nature, and sequence of the molecular causes leading to amyloidosis may differ between the different variants [8]. In addition, fibril accumulation and toxicity vary between different mutations, resulting in different expressions of the disease and in different responses to treatment [8,9,10].

The instability of the TTR tetramers is related to different events [11]. Mutations result in destabilization and dissociation of TTR tetramers into unfolded monomers and dimers; these intermediate products may undergo further unfolding or partial refolding [11,12]. Mutations can also make the protein susceptible to thermodynamic instability that promotes the formation of insoluble intracellular, potentially toxic aggregates [5]. It is unclear when the toxic varieties are formed and if toxicity varies among different amyloid fibrils. Mutations may be responsible for tissue adverse responses resulting from intracellular or extracellular conditions, or fromtheir different structural or kinetic variants [11].

Structural changes in amyloidogenic TTR have been difficult to identify through X-ray crystallography; however, novel studies with nuclear magnetic resonance spectroscopy have revealed small chemical shifts to the backbone structure of mutated and WT-TTR [8]. These changes are significant to the TTR ground state and the most significant changes were found in proximity to the site of the mutation [8]. These subtle structural differences are just one consequence of the mutation, and different TTR variants might experience diverse structural deformation pathways, resulting in atypical responses to the cellular conditions or proteolytic stress [8,9,10]. The underlying molecular mechanisms could also be responsible for the tropisms to specific organs, the age at onset, or the disparities in the prognosis of the disease.

## 4. Endoplasmic Reticulum (ER) and Quality Control System (QCS) of the Cell and ATTR

Folding of proteins is assisted by molecular chaperones, which are key components of the ER QCS, a system that continuously performs surveillance of newly synthesized proteins employing a variety of strategies, which include the action of members of the major molecular chaperone families [13]. Molecular chaperones are ER-resident proteins that bind to and stabilize unstable conformers to promote correct folding, assembly of the substrate polypeptide, oligomerization and thereafter traffic to their destination [14,15,16,17]. The presence of non-native determinants indicating that the protein has not achieved a conformation compatible with ER export (e.g., exposure of hydrophobic shapes or amino amino acid sequences, unpaired cysteines or immature glycans) will result in ER retention by ER chaperones (e.g., BiP and GPR-94) of the QCS [14,18,19]. Irreversible misfolding results in retro-translocation of the defective protein across the ER and degradation by the ER-associated degradation (ERAD) proteasome system, which promote clearance of misfolded proteins to restore proteostasis [20,21]. In the case of some thermodynamically and kinetically unstable, highly-aggregation-prone TTR variants, recognition by the ER quality control pathways in the liver targets these variants (e.g., D18G and A25T TTRs) for degradation through ERAD; degradation of these highly unstable variants protects against severe early-onset systemic amyloidosis as their release from the liver is reduced [22,23,24]. In contrast, moderately unstable, but still amyloidogenic TTR variants (e.g., L559 TTR), may escape the ER QCS of the liver and be secreted at levels suitable to develop amyloidosis [25,26]. This does not mean that the ER QCS of the cell is inefficient to clear amyloidogenic TTR, but rather that the severity of the conformational alteration dictates the efficiency of the QCS to recognize and limit, or alternatively to promote the trafficking of the misfolded monomer or tetramer protein to the secretory pathway.

If restoration of the proteostasis network fails, unfolded and misfolded proteins (e.g., TTR monomers) accumulate in the ER resulting in prolonged stress [20,21], triggering the so-called unfolded protein response (UPR). The UPR is a regulatory mechanism that protect the ER from the overload of aberrant proteins, reducing the synthesis and translocation of new proteins while promoting that of others (e.g., transcription factors associated with UPR, molecular chaperones, degradation factors, and folding enzymes) [27]. As a result of the activation of the UPR, the folding capacity of the ER and the degradation of abnormally folded proteins increase, reducing ER stress. Nevertheless, if the ER QCS or UPR are impaired as a result, for example, of age (i.e., in the case of senile systemic ATTR), the secretion of amyloidogenic proteins may increase [28,29,30]. In this vein, several molecules involved in amyloid-forming disorders (e.g., presenilin 1 in Alzheimer’s disease or parkin in juvenile parkinsonism) may promote or inhibit various steps in the UPR [31]. In fact, it has been shown that the ER chaperone BiP binds the amyloid precursor protein in healthy cells, thereby limiting production of β-amyloid [32]. Mutations in presenilin 1 associated with familial Alzheimer’s disease may alter the transmembrane kinase and endoribonuclease sensor IRE1 (an UPR signaling pathway [33]) and reduce BiP levels, leading to ER stress and apoptosis or alternatively to enhanced β-amyloid production. Whether a similar situation occurs in the case of some ATTR variants, has not been explored in detail.

A complete understanding of the molecular mechanisms subserving the development of TTR amyloidosis will provide useful insights into the heterogenic manifestations of this misfolding disease. This knowledge could also aid in designing and maximizing new therapeutic strategies and drugs or implementing tailored approaches for specific subpopulations of patients suffering TTR amyloidosis. In this regard, activation of the protective UPR via promoting activation of its associated signaling factors might represent a potential therapeutic target for ATTR.

## 5. Amyloid Formation

Mutations impact the stability of the intermolecular contacts in the TTR tetramer region [11]. The dissociation of the tetramer into monomers that misfold is the critical step towards ATTR amyloidosis [2]. The tetramer can also be dissociated into TTR dimers forming intermediate products [4]. Chemical shifts under certain conditions destabilize the TTR protomer and facilitate local unfolding into monomers or dimers. After dissociation, the misfolded monomers or dimers are deformed into non-native oligomers to produce amyloid fibrils [10].

### 5.1. Dimers

In the pathway from tetramer dissociation to oligomer formation, TTR dimers act as building blocks. Although dimeric states may be intermediates from tetramers to monomers, they can also be formed after the monomerization of TTR. Dimeric species only maintain the hydrogen-bonded interface between the dimers A and C and B and D, and not the hydrophobic interface [12]. Dimer intermediates are prone to further aggregation and amyloid formation after complete denaturation [4].

### 5.2. Oligomers

Annular oligomers are composed of double stacks of octamers that tend to convert into spheroidal oligomers consisting of 8–18 monomers [34]. Aggregates of TTR mutant oligomers exhibit higher molecular weight than wild-type aggregates [4]. There is a good correlation between the rates of aggregation of TTR in vitro and the extent or severity of the disease phenotype [4]. Oligomers, rather than mature amyloid fibrils, seem to exert the toxic effect responsible for cell damage of the Schwann cell [2]. Also, TTR oligomers tend to be more evident in late rather than early-onset patients [4].

### 5.3. Protofibrils and Fibrils

Protofibrils are linear fibrillar intermediates formed from soluble oligomers [2]. Intermediate dimers or oligomers increase in size and are converted into more prominent protofibril aggregates in a nucleation-dependent manner [35]. Another mechanism is the polymerization of conformationally distorted tetramers, not dependent on the formation of unfolded monomers as aggregation intermediates [5,35]. The self-assembly of products of tetramer dissociation into pre-fibrillar aggregates is a crucial step in the formation and aggregation of amyloid. Regardless of the nature of the amyloidogenic intermediates, the assembly of fibrils is a continuum in the formation of amyloid deposits. A detailed analysis has shown mature TTR fibrils composed of a complex-linear arrangement of TTR protofibrils, heparan sulfate, and chondroitin sulfate proteoglycans [4], suggesting regional and tissue-specific differences in amyloid deposits from different TTR variants.

### 5.4. Seeding and Nucleation

Destabilized intermediates may serve a dual purpose as natural soluble, toxic oligomeric compounds or as seeds for further polymerization and aggregation. Seeding relies on the addition of preformed compounds, while nucleation implies the birth of new ones. Many oligomers have been identified in the early stages of fibrillation, serving as a precursor of amyloid nucleation [34]. Electron microscopic studies have shown amorphous electron-dense materials around microvessels and the subperineural space in addition to speckles or fine fibrillar-structures within the periphery, which may be the core of the amyloid fibrils [2]. Extended or shortened loops create new interactions at potential amyloid packing sites in which distorted tetramers join to form TTR amyloid fibers [7,8]. In addition to the toxic oligomers, the amyloid fibrils per se may also work as a seed to facilitate TTR fibrilization, as seen in ex-vivo cardiac ATTR amyloidosis [36]. Cardiac-derived ATTR seeds accelerate fibril formation of wild type and monomeric TTR in amyloid fibrils extracted from autopsied and explanted hearts of ATTR patients [36].

### 5.5. Milieu Factors

TTR amyloidosis may be facilitated by local mechanisms operating under different conditions, including pH and temperature. For example, WT-TTR cardiac amyloidosis is solely composed of WT-TTR, and ATTTv amyloid deposits are mostly TTR variants. Still, after cardiac transplantation in patients with ATTv amyloidosis, the amyloid deposition progresses utilizing wild-type TTR [2].

Tissue damage is also dependent on the type of fibril. For example, mature long amyloid fibers seem to pull surrounding tissues in the cases with long and thick amyloid fibrils; in contrast, the short and fine amyloid fibrils of the late-onset or non-endemic areas do not have traction on neighboring tissues [2]. This may have implications in the proportion of damage in small and large fiber nerves [2].

Finally, disruption of the blood-nerve barrier of endothelial cells in endoneurial microvessel due to inflammation has been observed in late-onset ATTRv amyloidosis [2].

### 5.6. Toxic Effects of Fibrils in Tissues

Misfolded proteins may exert harmful effects by interacting with cells [4]. Amyloid proteins can provoke calcium entry in cells when embedded into the plasma membrane, forming pore-like structures with consequent aberrant ion conductance. Different aggregates may have toxic effects on cells, thereby contributing to the distinct pathogenesis of TTR mutations. TTR aggregates may bind directly to cationic channels of the dorsal root ganglia, causing a conformational change, leading to channel gating [4]. TTR can also bind to lipid membrane components altering cell membrane fluidity and triggering the opening of TRPM8 prototypic thermosensitive ion channels in dorsal root ganglia sensory neurons [4]. Moreover, the misfolded TTR monomer may cause inflammation by revealing epitopes that are normally hidden within the TTR tetramer to the immune system and thereby initiate antibody formation against misfolded TTR.

### 5.7. Dissociation, Misfolding, and Proteolysis

Numerous TTR mutations destabilize tetrameric TTR to misfold in different ways [10]. Proteolysis sensitivity studies suggest that over the pH range 5.1 to 3.9, the C-strand-loop-D-strand portion of TTR becomes disordered and moves away from the core of the β-sandwich fold to form a monomeric amyloidogenic intermediate [5]. However, different structural perturbations have been found in various mutations. For example, the L55P mutation induces substantial structural perturbations in both the DAGH and CBEF β-sheets, whereas the V30M perturbs the CBEF sheet primarily [8]. While monomeric intermediate products remain the primary mechanisms for protofibril formation, the natively folded structure may not be significantly altered in some species, albeit it can be susceptible to thermodynamics that can favor the appearance of the amyloidogenic intermediates [5].

Proteolysis of TTR is another proposed mechanism that facilitates misfolded monomers’ formation. Some studies suggest that the initial fibrillization of specific TTR mutants involve the proteolysis of the full-length TTR derived from the exposure of proteolytic sites of the protein, leading to the formation of truncated species. After proteolysis, TTR fragments tend to be more amyloidogenic and aggregate with complete proteins to form amyloid fibrils. Different proteases have been implicated in the production of TTR fragments, such as plasmin, which showed TTR proteolysis on WT-TTR and amyloidogenic variants, or subtilisin, which is a serine protease from a nonpathogenic microbe in the gut, *Bacillus subtilis* [37]. The increased gut permeability occurring during aging could increase subtilisin levels into the human plasma, contributing to the pathogenesis of WT-cardiac amyloidosis [7,34,37].

Other mechanisms of amyloid formation are described in Table 1.

## 6. Diagnostic Methods

Congo red staining with polarization microscopy is the gold standard method for detection of amyloid deposits in tissues [38]. Marked variations in the affinity for Congo red staining is related to the affinity of the C-terminal ATTR fragments for the Congo red staining. Type B fibrils are characterized by a strong affinity and a glittering appearance when visualized under polarized light, whereas type A have a weak affinity and absence of glittering birefringence [3]. Identification of the amyloid precursor protein is performed by immunohistochemistry, immunblotting, or mass spectrometry [38]. Sometimes, amyloid deposits are unevenly distributed or too small to be detected with these conventional techniques [3].

Magnetic resonance imaging (MRI) with gadolinium enhancement has high sensitivity and specificity for amyloid cardiomyopathy. Tc-pyrophosphate (^99m^Tc-PYP) and Tc-diphophono-1,2-propanodicarbocylic acid (^99m^Tc-DPD) scintigraphy and SPECT have proven to be good diagnostic tools, particularly for cardiac amyloid deposits. Tc-DPD has lower affinity for type B ATTR deposits [3], while ^99m^Tc-PYP is dependent on the presence of C-terminal ATTR fragments. DPD and PYP uptake in ATTR amyloid is significantly higher than in light-chain (AL) amyloid [39,40].

Mass spectroscopy is typically performed only in specialized centers with expertise, and do not provide sufficient information about the extent or distribution of amyloidosis, disease progression, or response to treatment, and in practice may cause delayed care [40]. Amplifying ATTR by exploiting the amyloid seeding approach is being considered. However, the main drawback for using these techniques is the energetically favorable monomer entropy [38]. New assays for detecting ATTR in tissues are being described, mostly involving antibodies, particularly antibody-binding nanofibrils (Ab-bNF) which have 20-fold higher binding capacity thanprotein A-Sepharose [38].

## 7. Novel Treatments for ATTR Amyloidosis

The first specific drug approved for ATTRv amyloidosis was tafamidis, a small molecule that prevents tetramer dissociation of serum TTR [1]. Tafamidis occupies the binding sites of thyroxine, binding to a central pocket in the tetramer like thyroxine thereby preventing its dissociation [3]. Tafamidis can reduce the progression of neuropathy and cardiomyopathy, and in the latter it can reduce all-cause mortality and cardiovascular-related hospitalizations. However, it has shown better results in trials for patients in the early state of early-onset disease and in patients with the V30M mutation, compared to patients in more advanced stages, or with late-onset disease, where type A fibrils predominate [3]. Acoramidis (formerly known as AG10), is a newer, more potent kinetic stabilizer of TTR. It has provedto be safe and effective in patients with cardiomyopathy, and two clinical trials are ongoing in subjects with cardiomyopathy and polyneuropathy [41].

Patisiran is a double-stranded siRNA that can reduce the synthesis of TTR. Formulated as a lipid nanoparticle for targeted delivery to hepatocytes, patisiran significantly changed the primary and secondary outcomes in patients with ATTRv amyloidosis compared with placebo [42]. Patisiran delivery system involves a non-viral delivery, composed of ionizable cationic lipids in a 100 nm diameter particle covered with a PEG-lipid monolayer with water-filled cavities containing nucleic acid molecules [39]. This system is very efficient to enter the hepatocytes due to natural liver accumulation and interaction with ApoE [39]. Non-coding RNA molecules act on specific mRNAs through short guide strands that recognize complementary bases in TTR RNA. Patisiran has shown persistent knock-down of TTR gene expression and decreased serum TTR levels to up to around 6 years of use, without significant side effects.

Likewise, inotersen was developed to reduce hepatic TTR protein synthesis and serum TTR levels through RNAse H-mediated degradation of mutant and wild-type TTR mRNAs following complementary base pairing [43]. Inotersen is a 2′-MOE-modified antisense oligonucleotide (ASO) that uses short synthetic single-stranded DNA molecules to alter the splicing process. Inotersen showed to stabilize or improve neuropathy symptoms and reduce TTR levels, but it did not change the cardiological outcomes significantly [44].

Epigallocatechin-3-gallate (EGCG), the most abundant catechin in green tea and related flavonoids, inhibits fibril formation from several amyloidogenic proteins in vitro by binding to an alternative binding site, between the two dimers [45,46]. Its clinical benefit was observed in small observational studies of patients with ATTRv and WT-ATTR cardiomyopathy [46,47,48,49,50]. Curcumin (diferuloylmethane), the major bioactive polyphenol of turmeric, strongly suppresses TTR fibril formation in vitro, either by stabilization of TTR tetramer or by generating nonfibrillar small intermediates that are innocuous to cultured neuronal cells [50].It has been proposed as a potential treatment of ATTR amyloidosis [50].

The long-term efficacy of therapy for ATTR amyloidosis may be related to the possible pathophysiological effect of the TTR molecule. An effort should be made to elucidate which and how many mechanisms may be involved in the natural history of each type of ATTR amyloidosis, as one or more mechanisms may be at play, but not all respond equally to all the strategies proposed [51].

## 8. Conclusions

Many crucial aspects of TTR amylogenesis remain unresolved, particularly the variations in fibril configuration, refolding, aggregation, and fibril formation according to the TTR variant. Correlations to the severity of the clinical outcome, age of presentation, organ-specificity or tropism, and treatment success are still pending. Characterization of TTR intermediate states, such as dimers and oligomers and their structural polymorphisms, will provide critical insights into TTR aggregation mechanisms and help identify tailored therapeutic interventions.

## Figures and Tables

**Figure 1 ijms-22-13158-f001:**
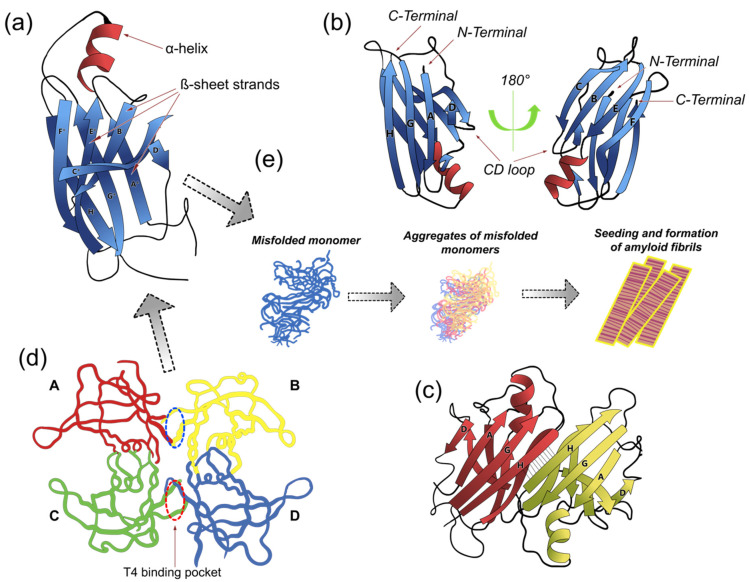
Transthyretin (TTR) conformational structures. TTR is a homotetrameric protein composed of four monomers of 127 amino acids each. Each monomer contains one small α-helix and eight β-strands (CBEF and DAGH) (**a**,**b**), which are arranged in a β-sandwich of two four-stranded β-sheets and one small α-helix found between β-strands E and F. TTR monomers interact via hydrogen bonds between the antiparallel, adjacent β-strands H-H’ and F-F’ to form a dimeric species (**c**). The two dimers (A-B and C-D) form the tetramer through hydrophobic contacts between the residues of the A and B, and G and H loops. The tetramer forms a central hydrophobic pocket (T4 channel) with two binding sites for hormones (red and blue ovals in (**d**)). (**e**) The TTR tetramer dissociates into dimers and lowest free-energy monomers more prone to form fibrils; mutant monomers misfold, aggregate, and subsequently form prefibrillar compounds and amyloid fibrils. In this scenario, tetramer dissociation into monomers is the rate-liming step of the aggregation reaction. Based on this model, several studies have focused on developing effective and selective therapeutic approaches (i.e., TTR ligands) aimed to prevent TTR dissociation and aggregation. (Model adapted from PDB code 1DVQ (https://pdbjbk1.pdbj.org/emnavi/quick.php?id=pdb-1dvq, (accessed on 5 October 2021)).

**Table 1 ijms-22-13158-t001:** Mechanisms of amyloid formation or stabilization.

**Amyloid Fibril Formation**
Formation of a a-sheet structures after tetramer dissociation Formation and aggregation of monomers, dimers or oligomers Seeding and nucleation Exposure of β-strand G (highly amyloidogenic) Conformational changes to an aggregation- prone state DAGH β-sheet flexibility with perturbation in an aggregation-prone state. Mutations in F residues Proteolysis
**Tetramer Stabilization Mechanisms**
Hydrogen-bond network Sidechain packing around the β-strands F and H β-strand H limits the exposure of the amyloidogenic β-strand G.

More than 130 mutations of the TTR gene are known to be pathogenic. Consequently, different mechanisms for amyloid formation have been described, some directly related to the mutation. Some regions of the TTR molecule are important for maintaining the tetrameric quaternary structure of TTR.

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
