# Peer review of "A Brief Journey through Protein Misfolding in Transthyretin Amyloidosis (ATTR Amyloidosis)"

_ijms, 2021, doi:10.3390/ijms222313158_

Round 1

Reviewer 1 Report

  1. Figures are not very precise and their content do not correspond to figure title

Figure 1

This figure is not informative, the graphic representations of the structures which are presented (from primary to quaternary) are not precise enough and scientifically sound.  The title does not correspond to figure content.

Figure 2

Again, the title does not correspond to the content of figure (the figure presents the amyloid forming pathways). The graphic representation is not very precise and description of used symbols (e.g. green line, S symbol) is missing.

  1. Tables

All tables should include the references for the presented information.

Table 1

Title should be shortened to the “Mechanisms of amyloid formation and stabilization”

Table 2

The title is not precise. It should be changed to “Comparison of TTR amyloid deposits”

Table 3

The Key-Points do not correspond to the content of the manuscript.

  1. The layout of the manuscript should be corrected. The sections: “Dimers”, “Oligomers, Protofibrils and Fibrils”, “Seeding and Nucleation”, “Milieu factors” and “Toxic effects of fibrils in tissues” should be included (as the subsections) into “Amyloid formation” section. As the associations of TTR with  inflammation are far greater than presented in the “Inflammation” section and deserve much more detailed description, the sentence: “The misfolded TTR monomer may reveal epitopes that are normally hidden within 267 the TTR tetramer to the immune system and thereby initiate antibody formation against 268 misfolded TTR. [3]” (lines 266-269) should be included into “Toxic effects of fibrils in tissues” section.

  1. The description of the structure of TTR (limes 56-61) should be corrected. The words: “is formed by” (line 58) should be changed to “of”. Otherwise, the sentence is incorrect for the definition of the secondary structure point of view.

  1. Throughout the manuscript there are many words that are used incorrectly, making their meaning unclear or incorrect, indicating poor English language skills.

Examples:

binding domain (line 54) should be exchanged with:  binding cavity/site

is built with (line 54) …is composed of

events (line 91)… causes

to (line 102) … in

proximal (line 103) … proximity to

misfold (line 163) … misfolding

deformed (line 166) … forming

prone (Table 1 line e) … aggregation-prone

a soluble oligomer (line 214) … soluble oligomers (plural form should be used here)

assemble (line 222) … assembly

interaction (line 235) … interactions (plural form should be used here)

less (line 284) … lower

amyloidosis (line 288) … amyloid

A-sepharosa (line 294) … A-sepharose

Sate (line 302) … stage

of TTR in the RNA (line 312) …in the RNA of TTR

lead (line 312) … leading!!!

Picture (line 333)… outcome

  1. Sometimes the information is incomplete and the words needed for correct understanding of the meaning are missing.

Examples:

Prognosis (line 25) … of the disease (?)

ß-sheet (line 53) … the word „content” is missing

AB and CD (line 65) … should be: A and B as well as C and D

Composition (line 76) … of what???

Numerous variants and single-site mutations (line 173) … single -site mutations and wild type molecules of TTR are TTR variants therefore this statement is incorrect

Proteolysis of the tertiary and quaternary TTR structure (line 184)… proteolysis can affect the protein structure but the tertiary or quaternary  structure cannot be proteolysed. Proteolysis results in the breaking of the covalent bond of the polypeptide chain.

susceptible (line 195) .. to what?

show a periodic structure of around 15-nm subunits of oligomers (line 215) …this statement is incomprehensible

additional (line 256) … to what?

Configuration (line 332) … of what ?

Fibril formation (line 332) … ability?

Ion strength (lines 332, 332) … of what?

Correlations (line 333) … to what?

Small fiber predominant axonal degeneration (Table 2) ….. axonal degeneration with predominant small fibers?

Severe nerve loss fiber (Table 2)  … this is not clear

Moderate nerve loss fibers (Table 2)   … this is not clear

Other toxic mechanisms of TTR amyloidosis (Table 3) … other than what???

Author Response

Dear Reviewer:  

Thank you for your thorough review. I am attaching a point-by-point answer to the suggestions. 

  1. Figures are not very precise, and their content does not correspond to the figure title 

       Figures were redesigned and merged into one.  

  1. Tables All tables should include the references for the presented information. Table 1 Title should be shortened to the “Mechanisms of amyloid formation and stabilization” Table 2 the title is not precise. It should be changed to “Comparison of TTR amyloid deposits” Table 3 The Key-Points do not correspond to the content of the manuscript. 

       We made changes to the tables. 

  1. The layout of the manuscript should be corrected. The sections: “Dimers”, “Oligomers, Protofibrils and Fibrils”, “Seeding and Nucleation”, “Milieu factors” and “Toxic effects of fibrils in tissues” should be included (as the subsections) into “Amyloid formation” section. As the associations of TTR with inflammation are far greater than presented in the “Inflammation” section and deserve much more detailed description, the sentence: “The misfolded TTR monomer may reveal epitopes that are normally hidden within 267 the TTR tetramer to the immune system and thereby initiate antibody formation against 268 misfolded TTR. [3]” (lines 266-269) should be included into “Toxic effects of fibrils in tissues” section. 

        The sections were moved, and the manuscript was reviewed to be consistent. 

  1. The description of the structure of TTR (limes 56-61) should be corrected. The words: “is formed by” (line 58) should be changed to “of”. Otherwise, the sentence is incorrect for the definition of the secondary structure point of view. 

       Suggested changes were made. 

  1. Throughout the manuscript there are many words that are used incorrectly, making their meaning unclear or incorrect, indicating poor English language skills 

        Changes were made accordingly, and the manuscript was reviewed to be consistent.  

  1. Sometimes the information is incomplete, and the words needed for correct understanding of the meaning are missing. 

        Phrases were completed or corrected.

Reviewer 2 Report

Although I found the review manuscript “A Brief Journey Through Protein misfolding in transthyretin amyloidosis (ATTR amyloidosis)” by Duarte and Ulloa-Aguirre somehow interesting, it fails to bring any novel perspective into the field of TTR amyloidosis. Indeed, this review, is line with many others recently published in the field in recent years.

Surprisingly, the authors also failed to acknowledge key discoveries in the molecular mechanisms underlying the TTR amyloid formation, with potential important therapeutic implications:

  1. Toxic TTR aggregation cascade (on-pathway) can be redirected by small molecules into the formation of non-toxic, amorphous TTR polymorphs that are innocuous to cells and tissues (off-pathway) (PMID: 21740906; PMID: 29124175; PMID: 30875761; PMID: 23069388).
  2. The central TTR pocket has been the target for many small-molecules stabilizers, including tafamidis, for the last 2 decades. This is very well-know and established. No novelty here. Nevertheless, it been previously described that binding of EGCG (and related flavonoids) to an alternative binding site at the surface of the molecule, between the two dimers, efficiently stabilizes the tetrameric fold, inhibiting its dissociation into toxic aggregate species (PMID: 20565072; PMID: 19861125; PMID: 22253829). These findings have great therapeutic interest, especially to TTR variants bearing mutations within the T4 binding pocket, where most previously reported TTR stabilizers are less effective.
  3. The beneficial effects of EGCG have been shown not only in preclinical models of disease, but most importantly in patients with cardiac TTR amyloidosis (PMID: 26673202; PMID: 22584381).

It is puzzling how the authors failed to mention all these relevant works made by well-known and well-established researchers in the TTR field, in a TTR amyloidosis review manuscript. Therefore, would strongly recommend the authors to acknowledge and further discuss these relevant findings in the revised form of the manuscript.

Author Response

We appreciate and thank the reviewer for the kind suggestions and we added these findings to the manuscript.

Round 2

Reviewer 1 Report

In the sentence” Transthyretin (ATTR) is a complex protein produced in the liver and choroid plexus that carries thyroxin and holo-retinol binding protein in serum” the abbreviation for transthyretin should be TTR. ATTR stands for the amyloidogenic form of TTR or amyloid of TTR. I would also recommend to omit the word “complex” and use thyroxine instead of thyroxin. In the next sentence the article “a” should be exchanged with article “an”.

In the sentence “ ATTRWT classically present with cardiomyopathy, carpal tunnel syndrome, and radiculopathy in older patients, usually above the age of 70.” the verb is missing. One can use the passive voice: “is present”.

In the sentence: ”This knowledge could also aid in designing new therapeutic strategies and drugs or implementing tailored approaches for specific subpopulations of patients suffering TTR amyloidosis.” suffering from TTR amyloidosis” should be used.

In the sentence: “The strands C-B-E-F are oriented orthogonally to the strands D-A-G-H, forming a prominent ß-barrel. (Figure 1C).” the expression “forming a prominent ß-barrel” should be removed. Otherwise the two different topologies are attributed to the same structure, as the following sentence describe it as a ß-sandwich (which is correct). Classic ß-barrel structure is attributed to RBP.

Figure 1 caption e) what does “free energy monomers” mean?

Does the sentence “So far, only 29 mutations have the kind of fibril composition examined.” was supposed to mean “So far, only for 29 mutations the type of fibril composition has been examined.”?

“In contrast, moderately unstable, but still amyloidogenic TTR variants (e.g. L559 TTR), may escape the ER QCS of the liver and be secreted at levels to develop amyloidosis.” …at the levels suitable to develop…

I do not understand the sentence: “Chemical shifts under certain conditions destabilize the TTR promoter and facilitate local unfolding into monomers or dimers” Did you mean protomers”?

The expression “natively folded but conformationally distorted tetramers,” is contradictory

The expression “„distorted, but intact” is contradictory

“Numerous TTR variants destabilize tetrameric TTR to misfold in different ways.” “mutations” should be used instead of “variants”

In the sentence: ”While monomeric inter-mediate products remain the primary mechanisms for protofibril formation, the native folded structure…. “ native folded should be changed to natively folded

“Other mechanisms of amyloid formation are described in Table 2. “ in fact, the mechanisms of amyloid formation are described in Table 1. Please note, that after this correction Table 2 will not be referred to in the manuscript.

“Type B is characterized by a strong affinity and a glittering appearance when visualized under polarized light…” should be: “Type B fibrils …”

The sentence: “However, the main drawback for using these techniques is the energetically favorable mono-mer entropy.” Is not clear. Did you mean that the monomer addition is energetically favorable?

protein A-seprharose should be corrected to protein A-Sepharose

Author Response

Dear Reviewer, 

Thank you for your through review of our manuscript. We addressed each one  and made the suggested changes. 

Best regards, 

alejandra. 

Reviewer 2 Report

The authors have partially addressed my comments. There are few final mandatory corrections to be made before acceptance for publication.

Page 8.

"Epigallocatechin-3-gallate (EGCG), the most abundant catechin in green tea and related flavinoids, inhibits fibril formation from several amyloidogenic proteins in vitro by binding to an alternative binding site, between the two dimers. [45] [46]."

  • Flavinoids is misspelled, it should be flavonoids.
  • Ref. 45 (Kristen et al., 2021) is misplaced here. The correct references supporting these key findings are Ref. 46 (Ferreira et al., 2026), and PMID: 19861125 and PMID: 20565072, as indicated in my original review report.

Author Response

Dear Reviewer, 

Thank you for your pertinent observation. Flavonoid was changed and references were added to new version of the manuscript.

Round 3

Reviewer 2 Report

The authors have successfully addressed my concerns, and I therefore recommend the revised version of the manuscript for publication.